# Enhancement of the Surface Properties on Polypropylene Film Using Side-Chain Crystalline Block Copolymers

**DOI:** 10.3390/polym12112736

**Published:** 2020-11-18

**Authors:** Sho Hirai, Patchiya Phanthong, Hikaru Okubo, Shigeru Yao

**Affiliations:** Research Institute for the Creation of Functional and Structural Materials, Fukuoka University, 8-19-1 Nanakuma, Jonan-ku, Fukuoka 814-0180, Japan; patchiya@fukuoka-u.ac.jp (P.P.); hokubo@fukuoka-u.ac.jp (H.O.); shyao@fukuoka-u.ac.jp (S.Y.)

**Keywords:** polypropylene, surface modification, adhesive properties, hydrophilicity, block copolymer

## Abstract

The consumption of polypropylene (PP) has significantly increased over that of other materials because of its light weight, easy molding, and high mechanical strength. However, the applications of PP are limited, owing to the lack of surface properties, especially with respect to adhesive properties and hydrophilicity. In this study, we developed a surface modification method for enhancing the adhesive properties and hydrophilicity on the PP surface using a side-chain crystalline block copolymer (SCCBC). This method was simple and involved the dipping of a PP film in a diluted SCCBC solution. The optimized modification conditions for enhancing the adhesive properties of PP were investigated. The results revealed that the adhesion strength of PP modified with the SCCBC of behenyl acrylate and 2-(tert-butylamino)ethyl methacrylate was enhanced to 2.00 N/mm (T-peel test) and 1.05 N/mm^2^ (tensile shear test). In addition, the hydrophilicity of PP modified with the SCCBC of behenyl acrylate and di(ethylene glycol)ethyl ether acrylate was enhanced to a water contact angle of 69 ± 4°. Surface analysis was also performed to elucidate a plausible mechanism for PP modification by the SCCBCs. This surface modification method is facile and enhances desirable properties for the wide application of PP.

## 1. Introduction

Plastic is one of the most used materials in daily life, owing to its light weight, low cost, and high mechanical and chemical properties. Since the mass production of plastic began in the 1950s, the world plastic production has significantly increased from 2 million tons in 1950 to 380 million tons in 2015, with an 8.4% increase in the production rate [1]. Many types of plastics with a variety of desirable properties have been developed and produced to date. Among them, polypropylene (PP) has been widely used in consumer products such as packaging materials [2], pipes [3], rope [4], and automobiles [5,6]. This is because, compared to other plastics, PP has exhibited superior properties including high chemical and heat resistances, easy molding, and high mechanical tensile and compression strengths [7]. Therefore, PP had been used in various fields spanning from the industrial scale to daily life and accounts for 17% of the total plastic production [1]. Recently, the application of PP as a separator for lithium-ion batteries and as a biomaterial for various applications has been explored to expand its usage [8,9,10]. However, because PP solely consists of alkane chains, it exhibits a low surface free energy. As a result, PP shows low hydrophilicity, poor adhesion properties, and low dyeability, which limits its applications.

Various modification methods have been developed to enhance the surface properties of PP. Indeed, PP polyfunctionalization was achieved by fabricating PP as a composite material using different techniques. For example, lamination, which was stacked by some amount of thin films, showed the different properties with the fabrication by only one film [11,12,13,14]. The other technique was polymer blending which was directly mixed with PP after adding some materials such as polyolefin elastomer [15], polyphenol [16], polyamide 6 [17], glass fiber [18], calcium carbonate [19], and montmorillonite [20]. A method in which PP was bonded with antioxidants by chemical reaction was also reported [21]. Although these methods have successfully improved the hydrophilicity, adhesive properties, and mechanical strength of PP, some drawbacks have also been encountered. Particularly, PP showed low compatibility with other materials, so it was difficult for the additives to be homogeneously dispersed. Therefore, not only did the modified products show insufficient desirable properties, but the natural properties of PP such as mechanical strength also decreased as the result of the slight difference in the modification conditions (i.e., added amount, component type, kneading time, and film thickness). As a result, these methods required a high level of experimental techniques and long processing times until the composite materials exhibited satisfactory properties.

On the other hand, surface modification of PP is attractive because of its simplicity and good repeatability. In this technique, it is not necessary to change the properties of the whole compound. Instead, the products are tailored according to the market requirement by modification at the PP surface to achieve the required properties. Surface modification of PP has been mainly achieved by chemical and physical treatments. In the former, a functional group, such as a hydroxy and carboxy group, was generated by using a nitric acid or ozone [22,23]. However, these chemical treatments require hazardous reagents to oxidize the highly chemical stabile PP surface. Therefore, the surface modified by chemical treatment is difficult to apply in a thin film or as a biomaterial. Hence, most surface modifications have been performed by physical treatments such as plasma [24,25], corona [26,27], and flame treatment [28,29]. These methods could successfully impart hydrophilicity and adhesion properties by scission of the PP alkane chain at the material surface to generate functional groups. However, these methods also had some disadvantages, including the requirement of special equipment and short-lived modification stability. Thin film modification by physical treatment is also difficult because the high energy of plasma irradiation is necessary. As a result, the mechanical strength of PP is significantly decreased owing to the extensive scission of the PP alkane chains. Moreover, these methods are limited by the type and shape of the substrates such as the porous membranes and the large size of the materials because the plasma cannot penetrate inside the porous membranes. From this viewpoint, some methods have been demonstrated to be mild and sufficient procedures by using a combination of chemical and physical treatments are required. These modifications were achieved in chemical reactions such as graft polymerization and oxidization on the PP surface using chemical reagents accompanied by UV irradiation [30,31,32]. These methods successfully imparted various properties to the PP, with long-term modification effects; however, they required complex techniques and special apparatus. Therefore, the development of facile modification techniques that enhance the adhesive properties of the PP surface is still challenging.

In our previous studies, the surface modification of polyethylene (PE) was successfully achieved by using a side-chain crystalline block copolymer (SCCBC). This comprised a block copolymer with a long alkyl chain and a functional group (amino or hydroxy group) as the respective side-chain crystalline and functional units. The result revealed that the side-chain crystalline unit was adsorbed on the PE surface because of the similar structures of the SCCBC side-chain crystalline unit and PE chain. In addition, the functional unit covered the PE surface, which resulted in the specific characteristics from the functional unit that were exhibited on the PE surface. This method could enhance the dispersibility, hydrophilicity, and adhesive properties of PE [33,34,35,36]. However, it was only applied to the PE surface because the side-chain crystalline units could only be adsorbed by PE, owing to their similar structures. In this current study, surface modification was developed on the PP surface by using two types of SCCBCs to enhance the adhesive properties and hydrophilicity of PP. The surface modification conditions, namely the concentration of SCCBC solution, solvent types, processing temperature, and dipping time, were varied to investigate the optimal conditions for improvement of the adhesive properties of the PP film. The adhesion strength was evaluated by the T-peel and tensile shear tests. In addition, surface analysis by atomic force microscopy (AFM), transmission electron microscopy (TEM), and Fourier transform infrared (FTIR) spectroscopy of the non-modified PP and PP modified with SCCBC was performed to elucidate the surface modification mechanism. Furthermore, the hydrophilicity enhancement of the PP modified with SCCBC was investigated. This study aimed at enhancing the surface properties of PP by using a facile method, to expand the usage of PP in a wide range of applications.

## 2. Materials and Methods

### 2.1. Material

All PP pellets: homopolymers (h-PP; Prime Polypro^TM^ J137G), random copolymers (r-PP; Prime Polypro^TM^ B241), and block copolymers (b-PP; Prime Polypro^TM^ B-150M), were purchased from Prime Polymer Co., Ltd., Tokyo, Japan. High-density polyethylene pellets (HDPE; FX201A) were purchased from Keiyo Polyethylene Co., Ltd., Tokyo, Japan. These PP and HDPE pellets were compressed at 210 °C (PP) and 180 °C (HDPE) for 2 min under 25 MPa by a hot compression machine IMC-180C (Imoto machinery Co., Ltd., Kyoto, Japan) to fabricate thin films of 0.5 mm thickness. Next, the thin films were cut into the dimensions 100.0 × 12.5 × 0.5 mm (length × width × thickness) for the T-peel tests and 100.0 × 25.0 × 0.5 mm for the tensile shear tests. Polyvinyl chloride (PVC) sheets (thickness: 0.5 mm) were purchased from Meiwa Gravure Co., Ltd., Osaka, Japan, while aluminum and copper metal sheets (thickness: 0.5 mm) were obtained from HIKARI Co., Ltd., Osaka, Japan. These substrates were cut to the same dimensions of the PP thin film. Aron Alpha^®^ 201, which comprised α-cyanoacrylate as its main component, was obtained from Toagosei Co., Ltd. (Tokyo, Japan) and was used as an instant glue. The stabilizer like a monomethyl ether hydroquinone contained in the monomers of behenyl acrylate (BHA, NOF Corp., Tokyo, Japan), 2-(tert-butylamino)ethyl methacrylate (TBAEMA, Sigma-Aldrich Co. LLC., St. Louis, MO, USA), and di(ethylene glycol)ethyl ether acrylate (DEEA, Sigma-Aldrich Co. LLC., St. Louis, MO, USA) were removed by inhibitor removers (Sigma-Aldrich Co. LLC., St. Louis, MO, USA) before use. 2-Methyl-2-[N-tert-butyl-N-(1-diethoxyphosphoryl-2,2-dimethylpropyl)aminoxy]propionic acid (BlocBuilder^®^ MA) was supplied by Alkema Inc., Colombes, France, and used as received. Octane, xylene, butyl acetate, decalin, benzonitrile, isopropanol, N,N-dimethylformamide (DMF), and methanol were purchased from FUJIFILM Wako Pure Chemical Corp., Osaka, Japan and used without further purification.

### 2.2. Synthesis of the SCCBCs

Two SCCBCs (BHA-TBAEMA and BHA-DEEA) were synthesized by nitroxide living radical polymerization according to our previously reported method [22]. BHA (5.0 g, 13.1 mmol), BlocBuilder^®^ MA (0.4 g, 1.0 mmol) and butyl acetate (5.7 mL) were added to a separable flask. The mixture was then deoxygenated by stirring under nitrogen atmosphere for 20 min and subsequently heated at 110 °C for 24 h. Next, a solution of butyl acetate (5.7 mL) and TBAEMA (5.0 g, 27.0 mmol) was added to the mixture and stirred at 110 °C for 24 h, after which the mixture was exposed to air and cooled down to room temperature prior to reprecipitation by methanol. The afforded product was then collected by filtration prior to drying under reduced pressure at room temperature for 24 h to obtain BHA-TBAEMA as a pale yellow solid. The weight-average molecular weight (Mw) and polydispersity index (PDI) of BHA-TBAEMA, which were determined by gel permeation chromatography (GPC, HLC-8320GPC; Tosoh Corp. Tokyo, Japan) and calculated using polystyrene as a standard, were 6.6 × 10^3^ g/mol and 1.2, respectively. BHA-DEEA was also prepared using a synthetic procedure similar to that of BHA-TBAEMA but with the functional group DEEA (5 g, 26.6 mmol) instead of TBAEMA. The afforded white BHA-DEEA solid of was obtained with 11.9 × 10^3^ g/mol of Mw and 1.4 of PDI. The chemical structure of SCCBCs are shown in Figure 1.

### 2.3. Surface Modification of the PP Film Using SCCBCs

The PP film surface was cleaned by washing with acetone and drying at room temperature. The diluted SCCBC solution (0.1 wt%) was prepared by the adding SCCBC solid (0.4 g) into xylene (400 g). Next, the PP film was dipped in the diluted SCCBC solution at 80 °C for 5 min. Subsequently, the modified PP film was dried at room temperature for one day in order to remove the excess of solvent.

To investigate the optimal conditions to improve the PP film adhesive properties, the modification conditions were varied with respect to concentration of SCCBC solution (0.01, 0.05, 0.10, 0.50 and 1.00 wt%), solvent types of the diluted SCCBC solution (octane, xylene, butyl acetate, decalin, benzonitrile, isopropanol, and DMF), dipping temperature (20, 40, 60, 80 and 100 °C), and dipping time (0.17, 1, 5, 10 and 30 min).

### 2.4. Characterization

The adhesive strengths of the non-modified PP film and PP films modified with the SCCBCs were evaluated by the T-peel test and tensile shear tests by using an EZ-LX compact table-top tester (Shimadzu Corp., Kyoto, Japan). Two pieces of the modified PP film were glued with an instant glue and attached with a bonded area of 12.5 × 75.0 mm (width × length) for the T-peel test and 12.5 × 25.0 mm (width × length) for the tensile shear test. The schematics of the test specimens are shown in Figure 2. The bonded PP films were cured at room temperature for 24 h and the averaged thickness of the bonded layer was 0.1 mm. Adhesive strength measurements were performed using a peeling rate of 1 mm/min for the T-peel test and 10 mm/min for the tensile shear test. All sample data were averaged from five tests. The surface topography was evaluated by atomic force microscopy (AFM) using a the JPK NanoWizard^®^ Sense AFM (Bruker Nano GmbH., Berlin, Germany). The PP films were observed in tapping mode by using a PPP-NCHAuD (NANOSENSORS™, Neuchâtel, Switzerland), which comprised a pyramidal cantilever with silicon coated by a metallic layer. Transmission electron microscopy (TEM) was performed by using a JEM-2100F (JEOL Ltd., Tokyo, Japan). The PP films were coated with an epoxy resin and cut into small pieces with a thickness of 100 nm using an ultramicrotome with a diamond knife at −120 °C. The small PP pieces were then stained with RuO_4_ for 15 min and the TEM measurements were performed with an accelerating voltage of 200 kV. Fourier transform infrared (FTIR) spectra were recorded by a Spectrum Two (PerkinElmer Inc., Waltham, MA, USA). The hydrophilicity was evaluated with a DropMaster DM-301 (Kyowa Interface Science Co., Ltd., Saitama, Japan) and the water contact angle was averaged from five testing samples.

## 3. Result and Discussion

### 3.1. Effect of the Surface Modification Conditions on the Adhesive Properties of PP Modified with BHA-TBAEMA

The effects of the surface modification conditions were investigated in order to enhance the adhesive properties of PP. The surface modification factors were varied with respect to the dipping temperature, dipping time, concentration, and types of solvent in the SCCBC solution. Figure 3a and Appendix A showed the adhesion strengths of the non-modified PP and PP modified with BHA-TBAEMA with different dipping temperatures (20, 40, 60, 80 and 100 °C) evaluated by the tensile shear and T-peel tests, respectively. In the case of the non-modified PP, the adhesion strength was 0.03 N/mm for the T-peel test and 0.18 N/mm^2^ for the tensile shear test, which implied that the neat PP surface lacked adhesion strength. The modified PP dipped in 0.1 wt% of BHA-TBAEMA in xylene for 10 min at 20 °C was also easily peeled off with an adhesive strength of 0.07 N/mm for the T-peel test and 0.46 N/mm^2^ for the tensile shear test. This adhesion strength was slightly enhanced when the dipping temperature of the PP modified with BHA-TBAEMA was increased to 40 °C. Notably, when the dipping temperature was further increased to 60 °C, the modified PP exhibited a significant increase in adhesion strength (1.97 N/mm for the T-peel test and 1.03 N/mm^2^ for the tensile shear test). This high adhesion strength was also observed when the PP was modified with the BHA-TBAEMA solution at dipping temperature of 80 and 100 °C. These results implied that the side-chain crystalline unit of SCCBC were strongly adsorbed onto the PP surface because side-chain crystalline unit could penetrate until the inner layer of PP and form cocrystals at high temperature. Based on an above investigation, we concluded that the optimized dipping temperature of the PP modified with BHA-TBAEMA solution was 80 °C, which afforded the highest adhesion strengths of 2.00 N/mm for the T-peel test and 1.04 N/mm^2^ for the tensile shear test.

The adhesion strength of PP modified with BHA-TBAEMA at different dipping times (0.17, 1, 5, 10 and 30 min) was next evaluated by tensile shear and T-peel testing (Figure 3b and Appendix A, respectively). When the dipping time was in the range 0.17–30 min, it did not affect the adhesion strength of the modified PP, and the adhesion strength averaged at approximately 2.00 N/mm for the T-peel test and 1.03 N/mm^2^ for the tensile shear test. Notably, all the test pieces were torn at the area of neat PP prior to peel-off. Therefore, this surface modification method enhanced the adhesive properties of PP in a short processing time. Indeed, the optimized dipping time of the PP in the BHA-TBAEMA solution was 5 min, which afforded the highest adhesion strength of 2.00 N/mm for the T-peel test and 1.05 N/mm^2^ for the tensile shear test.

The effect of the concentration of BHA-TBAEMA in xylene solution on the adhesion strength of PP is exhibited in Figure 3c and Appendix A. The concentration was varied at 0.01, 0.05, 0.10, 0.50 and 1.00 wt% of BHA-TBAEMA in xylene. At 0.01 wt%, the adhesion strength of the modified PP was slightly higher than that of the non-modified PP. These results considered that the adhesive properties on the PP surface were an insufficient enhancement owing to the lack of BHA-TBAEMA to modify the whole PP surface. A significant increase in adhesion strength was detected in the concentration range 0.05–0.10 wt%, reaching values of 2.00 N/mm for the T-peel test and 1.05 N/mm^2^ for the tensile shear at 0.10 wt%. However, the adhesion strength decreased with a further increase in concentration to 0.50 wt% and decreased sharply when the concentration exceeded 0.50 wt%. This result implied that the functional unit did not work well because excess BHA-TBAEMA adsorbed onto the PP surface to form the multilayer of copolymer. Thus, we concluded that the optimal concentration of BHA-TBAEMA in xylene was 0.10 wt%.

The adhesion strengths of the non-modified PP and PP modified with BHA-TBAEMA in different types of solvent were also evaluated, as shown in Figure 3d and Appendix A. The adhesion strength was significantly enhanced when the PP was modified with BHA-TBAEMA in octane and xylene as the solvents, while butyl acetate and decalin only afforded a slight increase in adhesion strength (approximately 1.00 N/mm for the T-peel test and 0.85 N/mm^2^ for tensile shear test). On the other hand, the use of benzonitrile, isopropanol, and DMF solvents did not affect the adhesion strength. This is because BHA-TBAEMA is more soluble in nonpolar solvents, such as octane and xylene, than the polar solvents, such as benzonitrile, isopropanol, and DMF. Thus, surface modification with BHA-TBAEMA in a nonpolar solvent was found to be suitable to enhance the adhesion strength of PP.

From the investigation of the surface modification conditions for the enhancement of the PP adhesive properties, the optimized conditions were 0.1 wt% BHA-TBAEMA in xylene solution at 80 °C for 5 min, whereby the PP adhesion strength was enhanced to 2.00 N/mm for the T-peel test and 1.05 N/mm^2^ for the tensile shear test. These optimized conditions were used in the subsequent characterization experiment.

Figure 4 shows the photos of the test pieces after testing by T-peel (Figure 4a) and tensile shear testing (Figure 4b). The non-modified PP films were easily peeled off at the adhesion area. On the other hand, the test pieces of PP modified with BHA-TBAEMA were torn at an area other than the bonded area. This implied that the adhesive properties of the PP surface were enhanced by modification with BHA-TBAEMA. As a result, the PP test pieces were broken at the neat PP area prior to peel-off at the bonded area.

To evaluate the adhesion strength of PP with other types of substrate, h-PP modified with BHA-TBAEMA was attached to another test piece of h-PP, r-PP, b-PP, or HDPE, also modified with BHA-TBAEMA under the same modification conditions. Both the T-peel test and tensile shear tests were then performed on these samples. The results were exhibited in Figure 5 and Appendix A. It can be found that the adhesion strength was enhanced not only for the test piece comprising h-PP attached with h-PP, but also for h-PP attached with r-PP, b-PP, and HDPE. In addition, all the test pieces were broken at the neat area prior to peel-off at the bonded area. The adhesive strength of the h-PP film modified with BHA-TBAEMA was further studied with other types of substrates which were non-modified aluminum, copper, and polyvinyl chloride (PVC) film. High adhesion strength was detected in the modified h-PP attached with the metal substrates such as aluminum (1.03 N/mm^2^) and copper (1.05 N/mm^2^), where the modified h-PP film was broken at the neat area out of the adhesives zone. The adhesion strength of the modified h-PP and non-modified PVC was 0.26 N/mm^2^, where the PVC film was elongated prior to peel-off from the bonded area. These results indicated that surface modification with BHA-TBAEMA successfully enhances the adhesive properties of PP with other plastic and metal substrates, which widens the range of applications for this material.

### 3.2. Surface Analysis of the Non-Modified PP and PP Modified with BHA-TBAEMA

The surface morphology and roughness average (Sa) of the non-modified PP and PP modified with BHA-TBAEMA were determined by AFM and the results are shown in Figure 6. The non-modified PP (Figure 6a) showed a surface with the Sa value of 1.8 nm, while the surface of the modified PP (Figure 6b) was rougher with the Sa value of 3.5 nm. This confirmed that surface modification by BHA-TBAEMA resulted in a higher roughness and changes in the PP surface, which led to an enhancement of the adhesive properties.

Figure 7 shows the TEM cross-sectional image of PP modified with BHA-TBAEMA. It can be found that the thin layer of BHA-TBAEMA was formed on the surface of PP with a thickness approximating 20 nm. Interestingly, some parts of the BHA-TBAEMA penetrated the PP layer. This implies that the long alkane chain crystalline unit of BHA-TBAEMA was not only adsorbed on the PP surface but also formed cocrystals and penetrated the inner PP layer.

The surface chemical composition of the non-modified PP, BHA-TBAEMA, and PP modified with BHA-TBAEMA were analyzed and FTIR spectra were displayed in Figure 8. The characteristic peaks of the non-modified PP were visible at 2951, 2915, 2837, 1455 and 1375 cm^−1^, which are typical alkane (C–H) peaks [37]. On the other hand, the modified PP showed new peaks that were assigned to the C–H at 2915 and 2850 cm^−1^, C=O ester group at 1733 cm^−1^, and C–O–C asymmetric stretching peak at 1160 cm^−1^. These new peaks corresponded to the characteristic peaks of BHA-TBAEMA. As a result, the surface of PP was successfully modified with BHA-TBAEMA via this surface modification method.

### 3.3. Plausible Mechanism for PP Modification with SCCBC

The schematic diagram of the plausible mechanism for the modification of PP with SCCBC is proposed in Figure 9. After dipping of the non-modified PP film into the SCCBC solution, the side-chain crystalline unit of the SCCBC was adsorbed on the PP film by van der Waals forces. Thus, cocrystals were formed between the side-chain crystalline unit of SCCBC and the PP alkane molecule. In addition, the functional unit of SCCBC covered the PP surface, thereby modifying the properties on the PP surface. As a result, the PP modified with BHA-TBAEMA exhibited the enhancement of adhesive properties, derived from the cocrystal of BHA on PP and the functional unit of TBAEMA on the PP surface.

The conditions of the surface modification method affected the mechanism. As shown in Figure 9, the difference in dipping temperature affected the adhesive properties of the PP modified with SCCBC. At low dipping temperatures, a weak adhesion strength was detected because of the small number of cocrystals between the side-chain crystalline unit of SCCBC and PP molecules. Most of the side-chain crystalline unit was adsorbed on the surface, and was easy to separate from the PP. On the other hand, at high dipping temperatures, the PP surface molecules could move easily, affecting the penetration of the side-chain crystalline unit of SCCBC into the inner layer of PP and forming as cocrystals. As a result, enhanced adhesive properties were exhibited on the PP surface modified with SCCBC in the types of BHA-TBAEMA.

### 3.4. Enhancement of the Hydrophilicity on the PP Surface by Modification with BHA-DEEA

Enhancement of the hydrophilicity on the PP surface is attractive for membrane and other applications. By using the surface modification in this study, SCCBC also improved the hydrophilicity of PP. Figure 10 shows the contact angle of a water droplet on non-modified PP and PP modified with BHA-DEEA. The hydrophobic surface of the former was detected at a water contact angle of 101 ± 1°. However, the water contact angle decreased to 69 ± 4° on the surface of the latter, implying that the hydrophilic properties of PP were enhanced. With the suitable selection of the functional group and modification conditions, SCCBC was able to enhance any desirable properties for the wide application of the PP surface.

## 4. Conclusions

The facile surface modification method for the enhancement of adhesive properties and hydrophilicity on the PP surface was developed. SCCBC comprising a side-chain crystalline unit and functional unit plays a key role in improving the surface properties of PP. The simple surface modification method comprised dipping of the PP film in a diluted solution of SCCBC. The optimized conditions were found to be 0.1 wt% BHA-TBAEMA in xylene solution at 80 °C for 5 min. As a result, the PP adhesion strength was enhanced to 2.00 N/mm for the T-peel test and 1.05 N/mm^2^ for the tensile shear test, in which the test pieces were broken at the neat area prior to peel-off at the bonded area. In addition, the adhesion strength was enhanced by h-PP attached to h-PP, other plastics, and metal substrates. From the surface characterization of PP modified with BHA-TBAEMA, it can be confirmed that BHA-TBAEMA adsorbed on the surface of PP. Moreover, it was deduced that the BHA side-chain crystalline unit penetrated the inner PP layer to form cocrystals, while the functional TBAEMA unit covered the PP surface, thereby enhancing the adhesive properties of the material. In addition, PP modified with BHA-DEEA exhibited the increase of hydrophilicity, with a water contact angle of 69 ± 4°. With the suitable selection of the functional group and modification conditions, SCCBC enhances any desirable properties to broaden the applications of the PP surface. In addition, this surface modification method was easily scaled up to an industrial scale because of its simple manipulation and its mild and wide range of modification conditions.

## Figures and Tables

**Figure 1 polymers-12-02736-f001:**
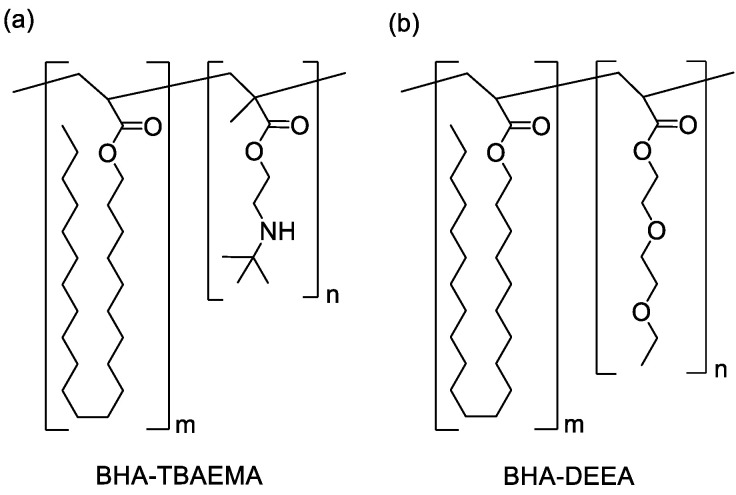
Chemical structure of SCCBCs (**a**) BHA-TBAEMA; (**b**) BHA-DEEA.

**Figure 2 polymers-12-02736-f002:**
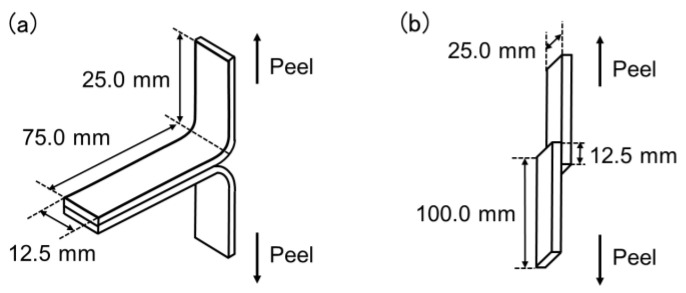
Schematics of the test specimen: (**a**) T-peel test; (**b**) tensile shear test.

**Figure 3 polymers-12-02736-f003:**
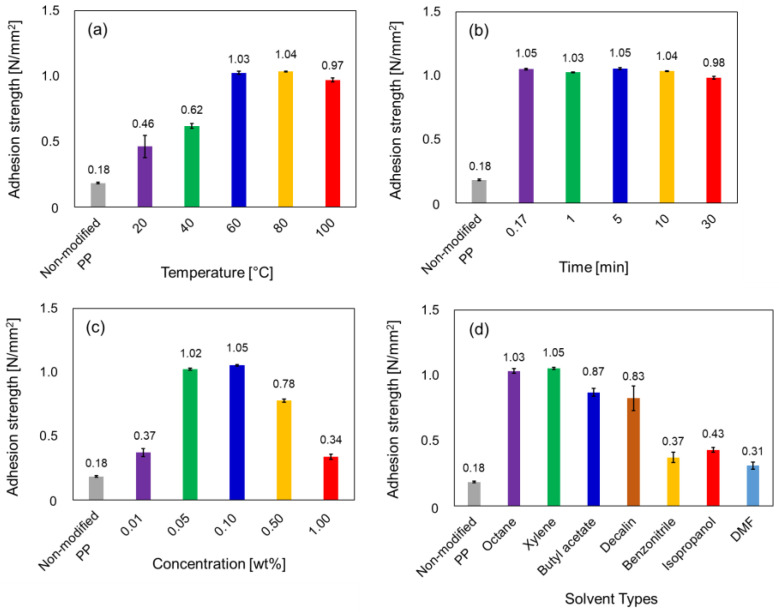
Adhesion strength of the non-modified PP and PP modified with the SCCBC evaluated by the tensile shear tests: (**a**) 0.1 wt% of BHA-TBAEMA in xylene for 10 min at different dipping temperatures; (**b**) 0.1 wt% of BHA-TBAEMA in xylene at 80 °C with different dipping times; (**c**) BHA-TBAEMA at 80 °C for 5 min at different concentrations of BHA-TBAEMA in xylene solution; (**d**) 0.1 wt% of BHA-TBAEMA in different solvent types at 80 °C for 5 min.

**Figure 4 polymers-12-02736-f004:**
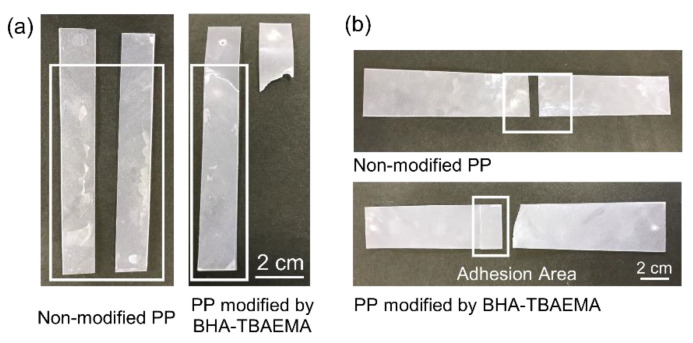
Photos of the test pieces after evaluation by adhesion measurement: (**a**) T-peel test; (**b**) tensile shear test.

**Figure 5 polymers-12-02736-f005:**
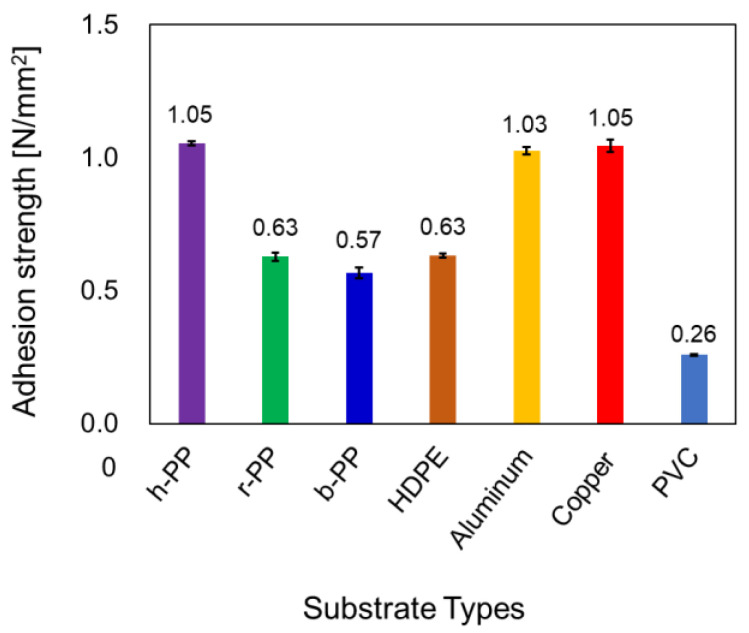
Adhesion strength of the modified h-PP attached with other substrates (h-PP, r-PP, b-PP, and HDPE), which were also modified with BHA-TBAEMA under the optimized conditions, and non-modified aluminum, copper, and PVC film. These results were evaluated by the tensile shear tests.

**Figure 6 polymers-12-02736-f006:**
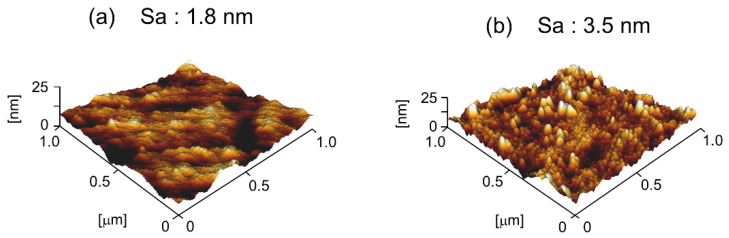
AFM topography images and surface roughness average parameters: (**a**) non-modified PP; (**b**) PP modified with BHA-TBAEMA.

**Figure 7 polymers-12-02736-f007:**
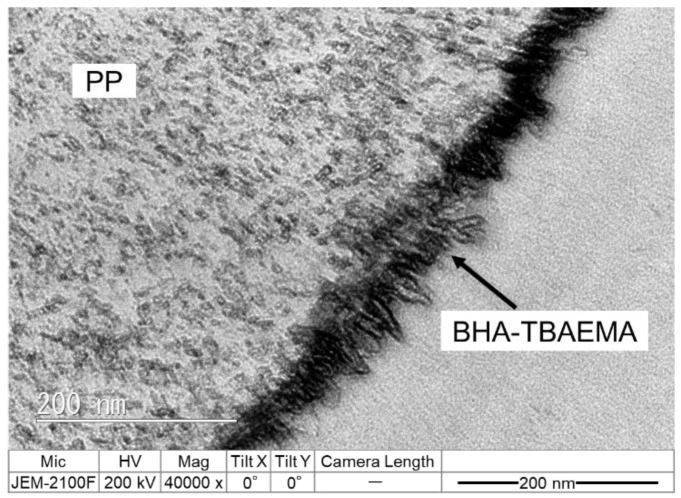
TEM cross-sectional image of PP modified with BHA-TBAEMA.

**Figure 8 polymers-12-02736-f008:**
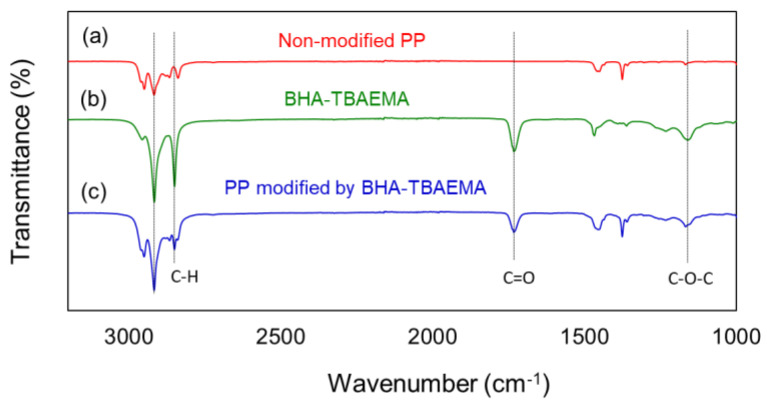
FTIR spectra: (**a**) non-modified PP; (**b**) BHA-TBAEMA; (**c**) PP modified with BHA-TBAEMA.

**Figure 9 polymers-12-02736-f009:**
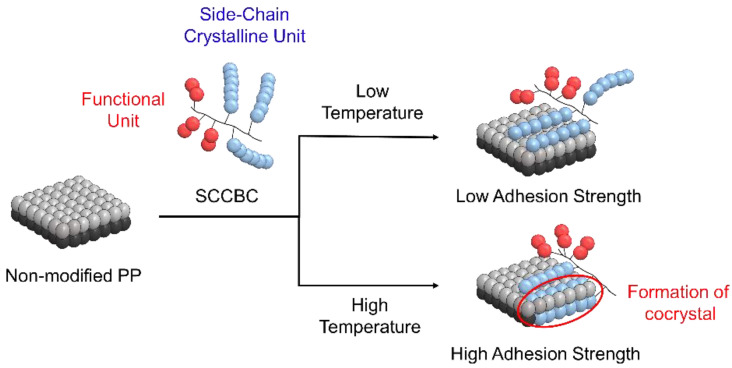
Schematic diagram of the plausible mechanism of PP modified with the SCCBC.

**Figure 10 polymers-12-02736-f010:**
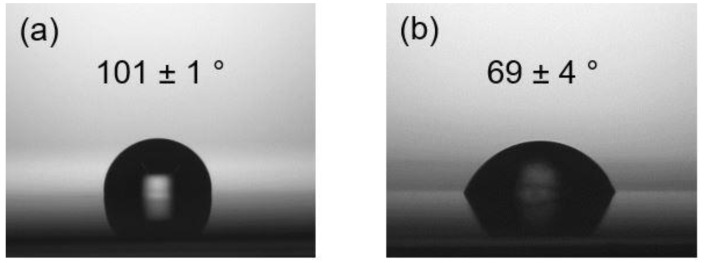
Water droplet images and the surface contact angle measurement: (**a**) non-modified PP; (**b**) PP modified with BHA-DEEA.

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
