# Peer review of "Enhancement of the Surface Properties on Polypropylene Film Using Side-Chain Crystalline Block Copolymers"

_polymers, 2020, doi:10.3390/polym12112736_

Round 1

Reviewer 1 Report

Authors present an interesting paper on the surface modification method for enhancing of adhesive properties and hydrophilicity on PP by using a side-chain crystalline block copolymer (SCCBC).

Authors developed a huge experimental work, by analying many factors as dipping temperature and time, solvent used for SCCBC solution, substrates, etc. They claimed for the optimal values of all the conditions, by just describing the results without any explanation about. I would appreciate an explanation of obtained results for all the parameters analyzed.

Moreover, even if the explanation obtained from AFM images seems to be feasible, I strongly recommend to improve the quality of those images, as they are not very clear.

In the same way, I recommend the reduction in the number of figures in which they analyzed the effect of different parameters. Some of them could be included as supporting information.

Finally, the language must be revised through the manuscript as many errors can be found. Some of them are the following:

104: All PP pellets which were homopolymer (h-PP; Prime PolyproTM J137G), random copolymer (r-104 PP; Prime PolyproTM B241) and block copolymer (b-PP; Prime PolyproTM B-150M) were purchased

44: drying in room temperature

273: and the results were showed in Figure 10

290: peaks of non-modified PP were appeared at 2951, 2915, 2837, 1455, 1375 which was derived from

308 the different in dipping temperature was affected to the adhesive properties

Reviewer 2 Report

  The manuscript entitled “Enhancement of surface properties on polypropylene film by using side-chain crystalline block copolymer” by Hirai and coworkers describes the use of a side-chain crystalline block copolymer (SCCBC) to enhance the adhesion and hydrophilicity of polypropylene (PP) films by immersing these films in dilute SCCBC solutions. The authors investigated the optimum amount of incorporated SCCBC that provided the best properties. Surface analysis was also performed to provide an explanation of the mechanism through which PP was modified by the SCCBCs.

  The water contact angles appear to be static contact angles. It may be helpful to supplement these static contact angles with either dynamic contact angles (such as advancing/receding contact angles) or alternatively sliding angle measurements (Gao, L. and McCarthy, T.J., Langmuir 2008, 24(17), 9183-9188).

  The writing of this manuscript could benefit from some polishing, as there are some sentences that seem to be unclear. I have included some suggestions in this regard below, but possibly additional proofreading beyond these suggestions may be advisable.

  This work will be of interest to researchers in various fields, such as polymer science, surface science, crystalline materials, materials science, and surface functionalization. The research was generally well-executed and the results were well-characterized. I believe that this manuscript is worthy f publication pending minor revisions, as outlined below.

Line 15: “was simple by the dipping of” can possibly be changed to “was simple and involved the dipping of”.

Line 29: “in 1950s” can be changed to “in the 1950s”.

Lines 31-32: The phrase “initiated and produced for serving of the desired properties.” Can possibly be changed to “developed and produced to fulfil a variety of roles” or “developed and produced with a variety of desirable properties”.

Line 37: A reference is needed for the phrase “and had been accounted for 17% of total plastic production”.

Lines 43-44: The phrase “processes of polyfunctionalized PP were performed by fabricated as a composite material with
traditional technique” is unclear.

Line 50: “were also occurred” can be changed to “were also encountered”.

Lines 51-52: “to be dispersed in homogeneous” can be changed to “to be homogeneously dispersed”.

Line 56: “until satisfaction properties were performed in composite materials” can possibly be changed to “until satisfactory properties were imparted to the composite materials” or “until the composite materials exhibited satisfactory properties”.

Lines 66-67: “could be successfully imparting of” could possibly be changed to “could successfully impart”.

Lines 82-83: “was succeeded to develop by using a” can be changed to “was successfully achieved by using a”.

Line 89: “could be easily increasing” could be changed to “could enhance the”.

Line 100: The phrase “This study expected to enhance the surface” is a little unclear.

Line 111: “for T-peel test” can be changed to “for T-peel tests”.

Line 111: “for tensile shear test” can be changed to “for tensile shear tests”.

Lines 118-119: The phrase “were removed stabilizer by an inhibitor removers” seems a little unclear.

Line 121: “France. and” can be changed to “France, and” .

Line 133: “to dried under” can be changed to “to drying under”.

Line 167: “cut into small piece” can possibly be changed to “cut into small pieces”.

Lines 183-184: The phrase “which implied to the lack of adhesion strength on neat PP surface” seems a little unclear. Possibly it could be changed to “which implied that the neat PP surface lacked adhesion strength”.

Line 192: The phrase “which affected to the highest adhesion strength” seems to be a little unclear.

Line 211: “enhance adhesive properties of” can be changed to “enhance the adhesive properties of”.

Line 212: The phrase “which affected to the” seems to be a little unclear.

Line 236: “more dissolved” can possibly be changed to “more soluble”.

Line 259: “substrate which were” can possibly be changed to “substrates which were”.

Lines 276-277: the phrase “affected to the higher roughness and the changes of PP surface which led to” seems to be unclear.

Line 289: “spectra was” can be changed to “spectra were” or “spectra are”.

Line 290: “were appeared at 2951, 2915, 2837, 1455, 1375” can be changed to “were visible at 2951, 2915, 2837, 1455, 1375 cm-1”. (with “appeared” changed to “visible” and the wavenumber units added).

Lines 294-295: The phrase “As a result, this surface modification method was successfully modified PP surface by BHA-TBAEMA” is a little unclear. Possibly it could be changed to “As a result, the surface of PP was successfully modified by BHA-TBAEMA via this surface modification method” or possibly “As a result, this surface modification method was successfully used to functionalize the surface of BHA-TBAEMA”.

Line 307: The sentence “The surface modification condition was affected to the difference of plausible mechanism” is unclear.

Line 308: “the different in” can be changed to “the difference in”.

Line 320: The phrase “SCCBC was also succeeded to improve” seems to be unclear.

Line 323: Error margins may be needed for the contact angles.

Lines 323 and 324: “which can be implied to the enhancement of hydrophilic properties of PP” can possibly be changed to “which implied that the hydrophilic properties of PP were enhanced”.

Round 2

Reviewer 1 Report

Authors have properly addressed the required questions and corrections, so the manuscript can be now published in its actual form